# PCH-2 collaborates with CMT-1 to proofread meiotic homolog interactions

**Stefani Giacopazzi**[ID]◉, **Daniel Vong**[ID]◉, **Alice Devigne**[ID], **Needhi Bhalla**[ID]*

Department of Molecular, Cell and Developmental Biology, University of California, Santa Cruz, Santa Cruz, California, United States of America

◉ These authors contributed equally to this work.

* nbhalla@ucsc.edu

## Abstract

The conserved ATPase, PCH-2/TRIP13, is required during both the spindle checkpoint and meiotic prophase. However, its specific role in regulating meiotic homolog pairing, synapsis and recombination has been enigmatic. Here, we report that this enzyme is required to proofread meiotic homolog interactions. We generated a mutant version of PCH-2 in *C. elegans* that binds ATP but cannot hydrolyze it: *pch-2^{E253Q}*. *In vitro*, this mutant can bind a known substrate but is unable to remodel it. This mutation results in some non-homologous synapsis and impaired crossover assurance. Surprisingly, worms with a null mutation in PCH-2's adapter protein, CMT-1, the ortholog of p31^{comet}, localize PCH-2 to meiotic chromosomes, exhibit non-homologous synapsis and lose crossover assurance. The similarity in phenotypes between *cmt-1* and *pch-2^{E253Q}* mutants suggest that PCH-2 can bind its meiotic substrates in the absence of CMT-1, in contrast to its role during the spindle checkpoint, but requires its adapter to hydrolyze ATP and remodel them.

**Data Availability Statement:** All relevant data are within the manuscript and its Supporting Information files.

**Funding:** This work was supported by the NIH (grant numbers R25GM104552 [D.V.] and

## Author summary

The production of sperm and eggs for sexual reproduction depends on meiosis. During this specialized cell division, homologous chromosomes pair, synapse and undergo meiotic recombination so that they are linked by at least one chiasma to promote their proper segregation. How homologous chromosomes ensure that these important interactions are with the correct partner is currently unknown. Here, we show that PCH-2 and its adapter protein, CMT-1, proofread homolog interactions to promote their fidelity and proper meiotic chromosome segregation.

## Introduction

Sexual reproduction relies on meiosis, the specialized cell division that generates haploid gametes, such as sperm and eggs, from diploid progenitors so that fertilization restores diploidy. To ensure that gametes inherit the correct number of chromosomes, meiotic chromosome segregation is exquisitely choreographed: Homologous chromosomes segregate in meiosis I and

R01GM097144 [N.B.]). Some strains were provided by the CGC, which is funded by NIH Office of Research Infrastructure Programs (P40 OD010440). The funders had no role in study design, data collection and analysis, decision to publish, or preparation of the manuscript.

**Competing interests:** The authors have declared that no competing interests exist.

sister chromatids segregate in meiosis II. Having an incorrect number of chromosomes, also called aneuploidy, is associated with infertility, miscarriages and birth defects, underscoring the importance of understanding this process to human health.

Events in meiotic prophase ensure proper chromosome segregation. During prophase, homologous chromosomes undergo progressively intimate interactions that culminate in synapsis and crossover recombination (reviewed in [1]). After homologs pair, a macromolecular complex, called the synaptonemal complex (SC), is assembled between them in a process called synapsis. Synapsis is a prerequisite for crossover recombination to generate the linkages, or chiasmata, between homologous chromosomes that direct meiotic chromosome segregation [2–8]. Defects in any of these events can result in chromosome missegregation during the meiotic divisions and gametes, and therefore embryos, with an incorrect number of chromosomes.

Coordinating the events of pairing, synapsis and crossover recombination is essential for their proper progression. For example, defects that uncouple pairing and synapsis can produce situations in which non-homologous chromosomes are inappropriately synapsed and unable to undergo crossover recombination. Mutations that produce non-homologous synapsis often identify mechanisms that are important for homolog pairing. For example, in maize, budding yeast and mice, where the initiation of meiotic recombination plays an integral role in promoting accurate homolog pairing and synapsis, mutations in genes involved in homologous recombination can produce non-homologous synapsis to varying degrees [9–17]. In *Drosophila* and *C. elegans*, two organisms in which correct pairing and synapsis can occur independent of recombination events [18, 19], other mechanisms ensure correct homologous synapsis, including integrity of meiotic chromosome axes [20–23], highly regulated chromosome movement [24, 25], centromere clustering [26] and specific histone modifications [27].

Despite these differences, in most organisms, the conserved AAA-ATPase PCH-2 (Pch2 in budding yeast, PCH2 in *Arabidopsis* and *Drosophila* and TRIP13 in mice) is crucial to coordinate events in meiotic prophase. *In vitro* and cytological experiments indicate that it does this by using the energy of ATP hydrolysis to remodel meiotic HORMADs [28–32], chromosomal proteins that are essential for pairing, synapsis, recombination and checkpoint function [20–22, 33–41]. HORMADs are a protein family defined by the ability of a domain, the HORMA domain, to adopt multiple conformations that specify protein function [42–44]. Meiotic HORMADs appear to adopt two conformations: a "closed" version, that forms upon binding a short peptide within other proteins [45]; and a more extended, or "unlocked," version [46]. The closed form of meiotic HORMADs assemble on meiotic chromosomes to drive pairing, synapsis and recombination [45]. The physiological relevance of the unlocked version is unknown.

Cytological experiments in budding yeast, plants and mice show that PCH-2 removes or redistributes meiotic HORMADs on chromosomes [28, 30–32]. This change in localization both contributes to and reflects the progression of meiotic prophase events [28, 30–32]. However, major events in meiotic prophase, such as pairing, synapsis initiation and recombination, precede the removal or redistribution of meiotic HORMADs, raising the question of how PCH-2 affects these events to produce the phenotypes reported in its absence. Indeed, the phenotypes associated with loss of *PCH2* are inconsistent with its only role being the redistribution of meiotic HORMADs [47–49]. In *C. elegans*, PCH-2 regulates meiotic prophase events but not the localization of meiotic HORMADs [50], indicating that 1) the removal or relocalization of meiotic HORMADs is not essential for PCH-2's effects on pairing, synapsis and recombination; and 2) this model organism is particularly relevant to better understand how PCH-2 regulates these events.

We previously showed that in the absence of PCH-2, homolog pairing, synapsis and recombination are accelerated and associated with an increase in meiotic defects, leading us to speculate that PCH-2 disassembles molecular intermediates between homologous chromosomes that underlie these events to ensure their fidelity [50]. Specifically, we hypothesized that these molecular intermediates involve meiotic HORMADs and PCH-2 remodels meiotic HORMADs to proofread these homolog interactions [50]. To test this model, we generated a mutant version of PCH-2 that binds ATP, and its substrates, but cannot hydrolyze ATP and therefore cannot remodel and release its substrates. We reasoned that if our hypothesis is correct, this mutant protein will "trap," and allow us to more easily visualize, inappropriate homolog interactions that are incorrectly stabilized. This is accurate: *pch-2*$^{E253Q}$ mutants delay homolog pairing and accelerate synapsis, producing non-homologous synapsis. Meiotic DNA repair occurs with normal kinetics but crossover assurance is lost, suggesting that crossover-specific recombination intermediates between incorrect partners may also be inappropriately stabilized. Surprisingly, loss of CMT-1, an adapter protein that is thought to be essential for PCH-2 to bind its substrates, phenotypically resembles *pch-2*$^{E253Q}$ mutants. Since PCH-2 localizes normally to meiotic chromosomes in the absence of CMT-1, these data indicate that CMT-1 is dispensable for PCH-2 to bind its meiotic substrates on chromosomes but is essential to hydrolyze ATP and remodel its substrates.

## Results

### PCH-2$^{E253Q}$ localizes normally during meiotic prophase

We introduced a mutation by CRISPR/Cas9 genomic editing [51, 52] in the Walker B motif of PCH-2, E253Q, that allows it to bind ATP, but not hydrolyze it. We designated this allele *pch-2(blt5)* but will refer to it as *pch-2*$^{E253Q}$ (*pch-2*$^{EQ}$ in Figures). In budding yeast, this mutation abolishes PCH-2's *in vivo* meiotic function [29]. *In vitro*, PCH-2$^{E253Q}$ retains high affinity nucleotide binding, forms stable hexamers and interacts with its substrates [53]. In meiotic nuclei, PCH-2$^{E253Q}$ localized normally to meiotic chromosomes (Fig 1A, grayscale images in S4A Fig), appearing as foci just prior to the entry into meiosis (also known as the transition zone) and co-localizing with the synaptonemal complex once synapsis had initiated (Fig 1B, grayscale images in S4B Fig). It was also removed from the synaptonemal complex in late pachytene, similar to wildtype PCH-2 (Fig 1C, grayscale images in S4C Fig).

### *pch-2*$^{E253Q}$ mutants delay pairing

Next, we analyzed pairing in *syp-1* mutants. SYP-1 is a component of the synaptonemal complex (SC), and *syp-1* mutants fail to assemble SC between paired homologs, allowing us to more easily visualize pairing intermediates in the absence of synapsis [6]. In *C. elegans*, *cis*-acting sites called pairing centers (PCs) are required for pairing and synapsis [54]. In the absence of synapsis, homologous chromosomes exhibit stable pairing of PC, but not non-PC, ends of chromosomes [54]. We monitored synapsis-independent pairing at PC sites by performing immunofluorescence against HIM-8, which binds PCs of X chromosomes [55] (Fig 2B). A single focus of HIM-8 indicates paired X chromosomes while two foci indicate unpaired X chromosomes. When we imaged stained germlines, we divided germlines into six equal sized zones (see cartoon in Fig 2A). Because meiotic nuclei are arranged spatiotemporally in the germline, this allows us to assess pairing as a function of meiotic progression.

*syp-1* mutants initiated pairing in zone 2, maintained pairing at the X chromosome PC in zones 3,4 and 5 and destabilized pairing in zone 6 (Fig 2C). As we previously reported [50], *syp-1;pch-2Δ* double mutants also initiated PC pairing in zone 2 but had significantly more nuclei with paired HIM-8 signals than *syp-1* single mutants (p value < 0.0001, two-tailed

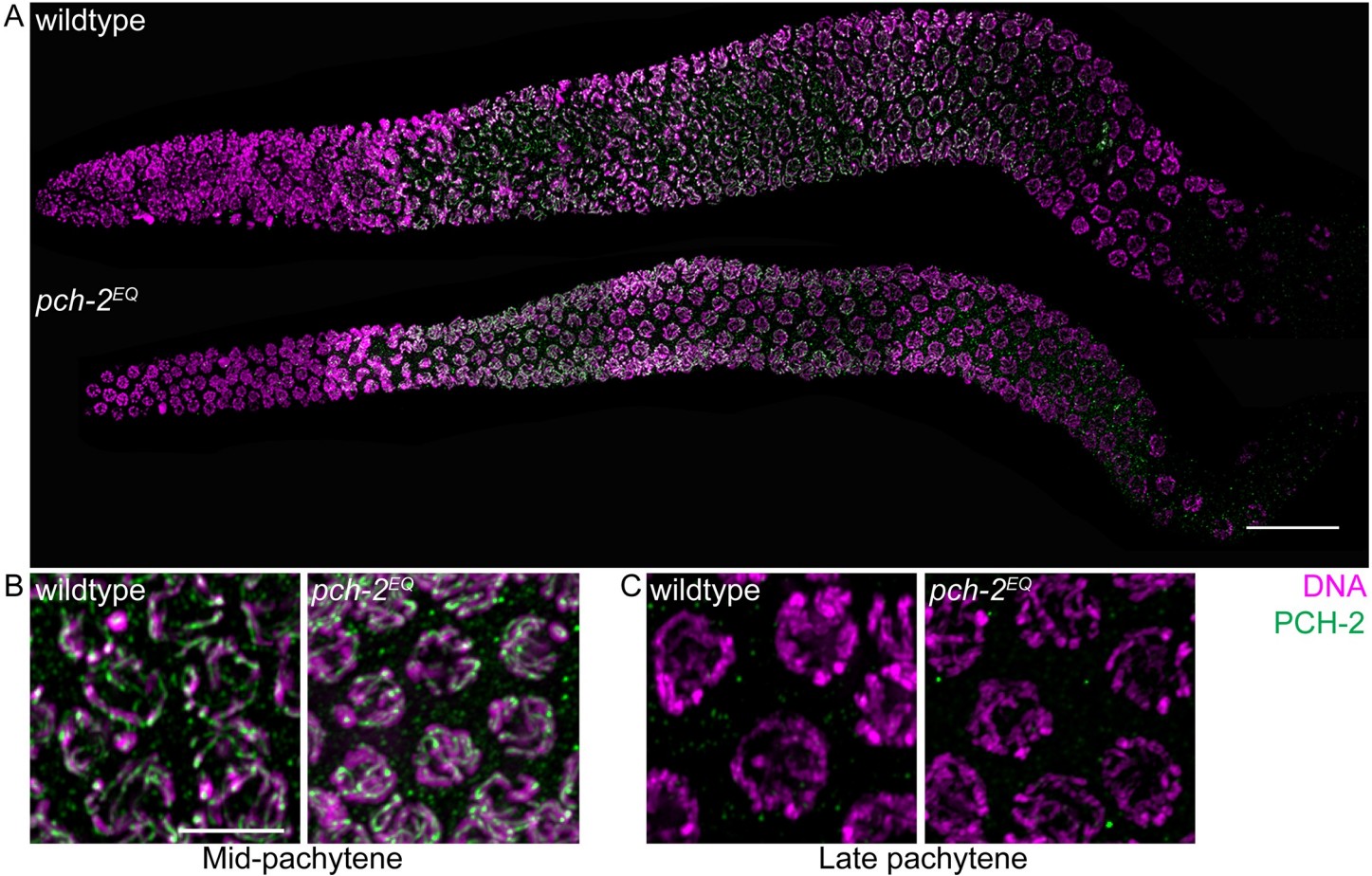

**Fig 1. PCH-2^E253Q localizes to meiotic chromosomes similar to wildtype PCH-2.** A. Whole germline images of PCH-2 and DAPI staining in a wildtype and *pch-2^E253Q* mutant germline. Scale bar indicates 20 microns. Meiotic nuclei in mid-pachytene (B) and late pachytene (C) stained with DAPI and antibodies against PCH-2 in wildtype animals and *pch-2^E253Q* mutants. Unless otherwise stated, all scale bars indicate 5 microns.

Fisher's exact test), indicating that pairing is accelerated in this background. By contrast, *syp-1; pch-2^E253Q* double mutants had significantly fewer nuclei with paired HIM-8 signals in zone 2 than both *syp-1* single (p value < 0.0001, two-tailed Fisher's exact test) and *syp-1;pch-2Δ* double mutants (p value < 0.0001, two-tailed Fisher's exact test). In later zones, all three genotypes closely resembled each other in that chromosomes are paired. However, in zone 4, significantly fewer X chromosomes were paired in *syp-1; pch-2^E253Q* double mutants than *syp-1* single mutants (p value = 0.0003, two-tailed Fisher's exact test). Thus, *syp-1; pch-2^E253Q* mutants delay and exhibit defects in pairing at PCs, consistent with our model that PCH-2 proofreads homolog pairing intermediates: If *pch-2^E253Q* mutants "trap" incorrect pairing intermediates, between non-homologous chromosomes, for example, this would affect pairing between homologous chromosomes.

### *pch-2^E253Q* mutants accelerate synapsis and produce non-homologous synapsis

Next, we assayed synapsis by monitoring the colocalization of two SC components, HTP-3 and SYP-1 [6, 54]. When HTP-3 and SYP-1 colocalize, chromosomes are synapsed while regions of HTP-3 devoid of SYP-1 are unsynapsed (see arrows in Fig 3A). We evaluated the

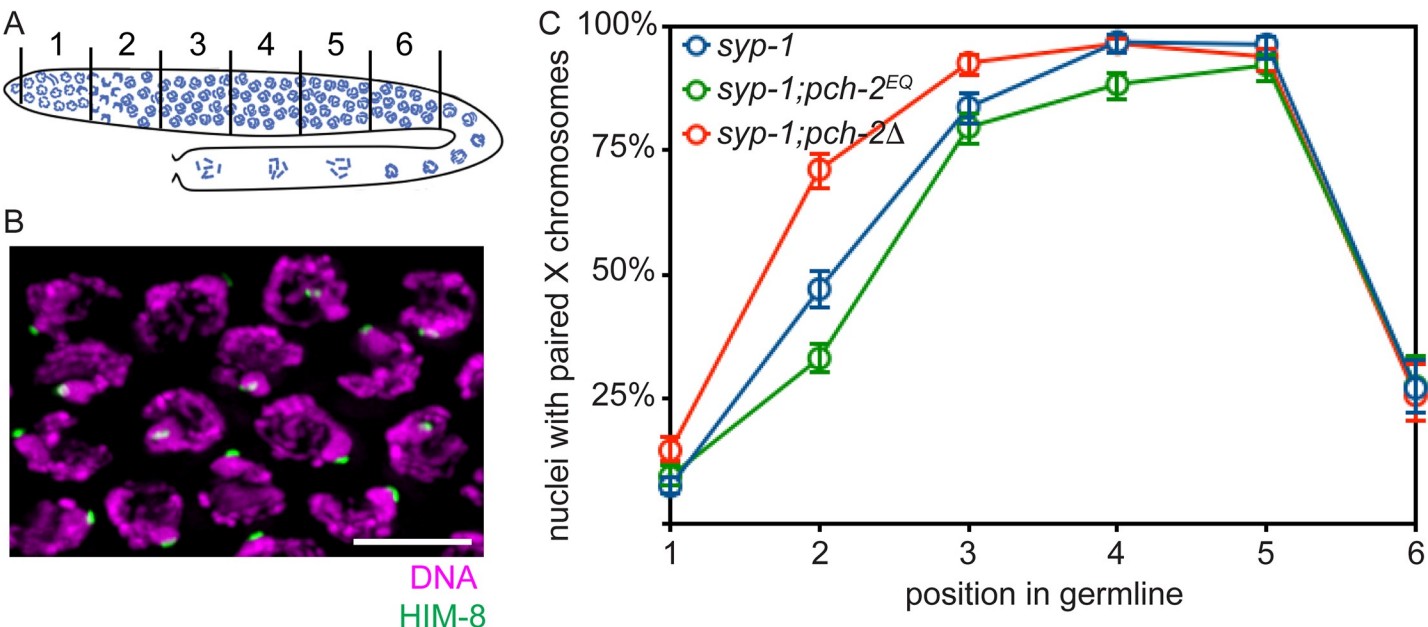

**Fig 2. Homolog pairing is delayed in *pch-2^{E253Q}* mutants.** A. Cartoon of *C. elegans* germline divided into six equivalent zones. B. Images of meiotic nuclei stained with DAPI and antibodies against HIM-8. C. Timecourse of X chromosome pairing in wildtype, *pch-2^{E253Q}* and *pch-2Δ* mutant germlines. Error bars indicate 95% confidence intervals.

percentage of nuclei that had completed synapsis as a function of meiotic progression (Fig 3B). In contrast to our analysis of pairing, synapsis in *pch-2^{E253Q}* mutants was accelerated, similar to but even more severely than *pch-2Δ* mutants (Fig 3B and [50]) (p value < 0.0001, two-tailed Fisher's exact test): More nuclei had completely synapsed chromosomes when synapsis initiates in zone 3 in *pch-2^{E253Q}* mutants than control germlines (Fig 3B) (p value < 0.0001, two-tailed Fisher's exact test). In contrast to *pch-2Δ* mutants, *pch-2^{E253Q}* mutants did not exhibit defects in SC disassembly in zone 6 (Fig 3B and [50]).

We reasoned that the delay in pairing, combined with the acceleration in synapsis, raised the possibility that these events, which are typically coupled, had been uncoupled. To test this possibility, we evaluated whether non-homologous synapsis occurs in *pch-2^{E253Q}* mutants. We stained wildtype, *pch-2Δ* and *pch-2^{E253Q}* mutant germlines with antibodies against SC components HTP-3 and SYP-1 to evaluate synapsis and HIM-8 to assess pairing. All nuclei in wildtype animals, *pch-2Δ*, and most nuclei in *pch-2^{E253Q}* mutants exhibited homologous synapsis (uncircled nuclei in Fig 3C). However, we identified meiotic nuclei in which all chromosomes were synapsed but contained two HIM-8 foci, indicating these chromosomes had synapsed with non-homologous partners (circled nuclei in Fig 3C). We also observed non-homologous synapsis when we monitored pairing of autosomes using antibodies against ZIM-2, a protein that binds an autosomal PC [56] (S1A Fig). When we quantified this defect, we detected non-homologous synapsis of X chromosomes in 2.5% of meiotic nuclei (Fig 3D) and non-homologous synapsis of Chromosome V in 6% of meiotic nuclei (S1B Fig). Unlike HIM-8 [55], ZIM-2 only stains meiotic chromosomes in a subset of meiotic nuclei [56]. Since fewer nuclei are positive for ZIM-2 than HIM-8 in all germlines, this explains the higher frequency of non-homologous synapsis that involves Chromosome V. *C. elegans* have six pairs of homologous chromosomes. Therefore, it's possible that as many as 18% of meiotic nuclei have at least one pair of chromosomes undergoing non-homologous synapsis in *pch-2^{E253Q}* mutants. However,

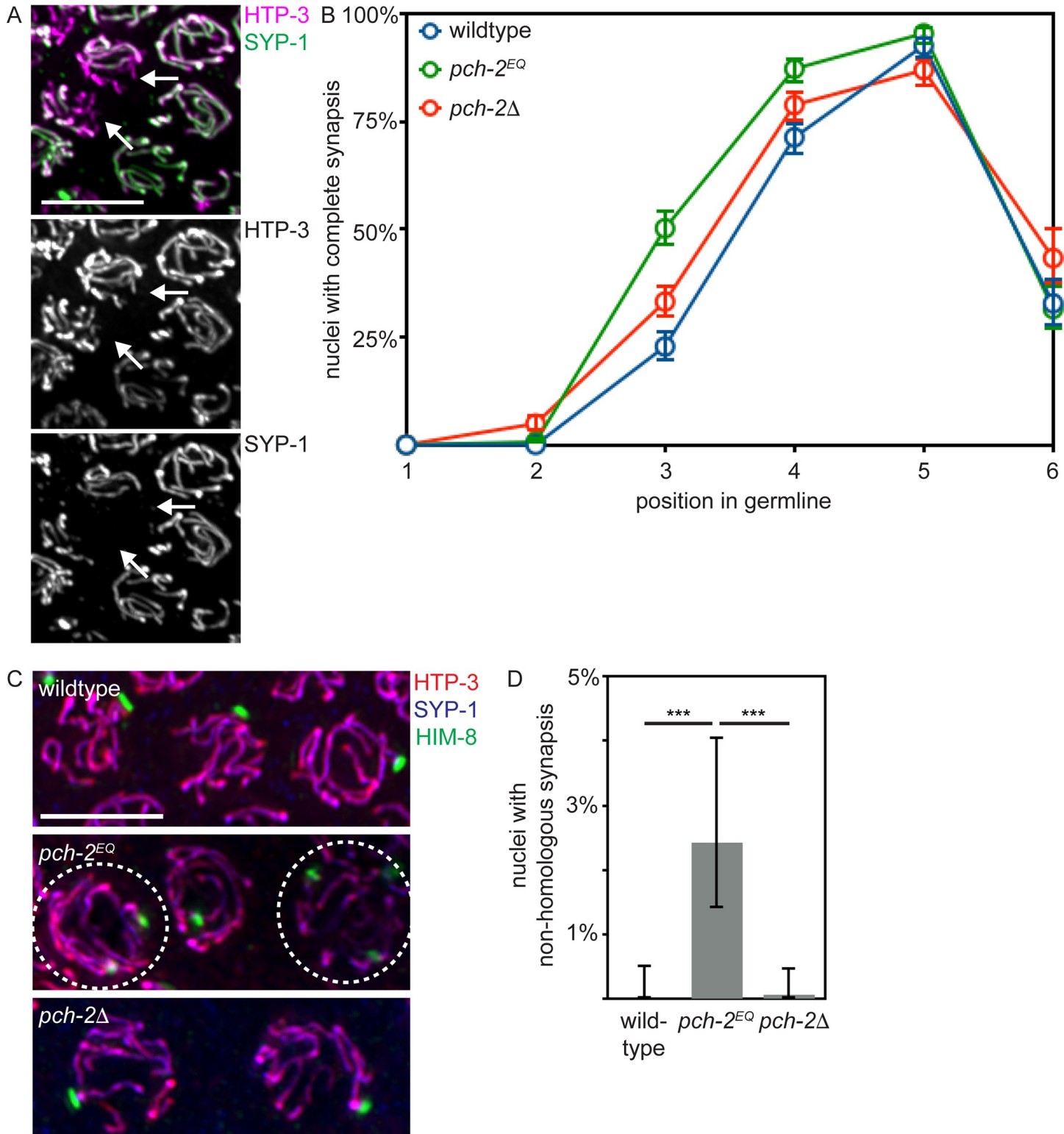

**Fig 3. Synapsis is accelerated in *pch-2^{E253Q}* mutants, producing non-homologous synapsis.** A. Images of meiotic nuclei stained with antibodies against HTP-3 and SYP-1. Color and grey scale images are provided. Arrows indicate unsynapsed chromosomes. B. Timecourse of synapsis in wildtype, *pch-2^{E253Q}* and *pch-2Δ* mutant germlines. C. Images of meiotic nuclei stained with antibodies against HTP-3, SYP-1 and HIM-8 in wildtype animals, *pch-2^{E253Q}* and *pch-2Δ* mutants. Circled nuclei have undergone non-homologous synapsis. D. Quantification of non-homologous synapsis wildtype animals, *pch-2^{E253Q}* and *pch-2Δ* mutants. All error bars indicate 95% confidence intervals. Significance was assessed by performing two-tailed Fisher exact tests. A *** indicates a p value < 0.0001.

it's also formally possible that some nuclei contain multiple pairs of chromosomes that are non-homologously synapsed and that 18% may be an overstatement. Unfortunately, we were unable to directly test this possibility due to technical limitations.

## *pch-2^{E253Q}* mutants lose crossover assurance

We then monitored meiotic DNA repair and the formation of crossovers. To assess DNA repair, we monitor the appearance and disappearance of RAD-51 from meiotic chromosomes as a function of meiotic progression (Fig 4A). RAD-51 is required for all meiotic DNA repair in *C. elegans* [57]. Its appearance on chromosomes signals the formation of double strand breaks (DSBs) that need to be repaired (see zones 3 and 4 in wildtype in Fig 4B) and its disappearance indicates the entry of DSBs into a DNA repair pathway (see zones 5 and 6 in wildtype in Fig 4B) [3]. The appearance and disappearance of RAD-51 on meiotic chromosomes in *pch-2^{E253Q}* mutants was indistinguishable from that in wildtype in zones 1–4. In zone 5, we detected fewer RAD-51 foci, suggesting that meiotic DNA repair may occur slightly more rapidly in *pch-2^{E253Q}* mutants (p value < 0.0001, two-tailed t- test). However, we observed an acceleration in DNA repair much earlier in *pch-2Δ* mutants (see zone 4, Fig 4B and [50] (p value < 0.0001, two-tailed t-test).

We assayed crossover formation in *pch-2^{E253Q}* mutants by visualizing GFP::COSA-1 foci formation in late meiotic prophase. GFP::COSA-1 cytologically marks putative crossovers and its appearance as robust foci is mechanistically associated with the process of crossover designation [58]. Almost all nuclei in wildtype worms had six GFP::COSA-1 foci per nucleus, one per homolog pair (Fig 4C, top, and 4D) [58]. In contrast, *pch-2^{E253Q}* mutants had a greater number of nuclei with five GFP::COSA-1 foci, similar to *pch-2Δ* mutants, indicating a defect in crossover assurance (Fig 4C, bottom, 4D and [50]). Given the *in vitro* behavior of the PCH-2^{E253Q} hexamer [53], this phenotype is likely because inappropriate crossover intermediates are stabilized at the expense of appropriate ones. Alternatively, these meiotic nuclei may result from the non-homologous synapsis we also observe in this mutant background. However, we do not observe a delay in meiotic DNA repair in *pch-2^{E253Q}* mutants (Fig 4B). Further, *pch-2^{E253Q}* mutants do not activate feedback mechanisms, as visualized by the extension of DSB-1 on meiotic chromosomes [59, 60] (S2A and S2B Fig). Therefore, we favor the first interpretation.

## CMT-1 is required for the synapsis checkpoint

During the spindle checkpoint, PCH-2 requires the presence of an adapter protein, CMT-1 (p31^{comet} in mammalian cells), to bind and remodel a HORMAD protein required for the spindle checkpoint response, Mad2 [53, 61, 62]. Based on a recent report that the rice ortholog of CMT-1 is an SC component and required for pairing, synapsis and double strand break formation [63], we tested whether *cmt-1* had a role in meiotic prophase by assessing whether it was required for meiotic prophase checkpoints in *C. elegans*. We introduced a null mutation of *cmt-1* into *syp-1* mutants. The absence of SC in *syp-1* mutants results in unsynapsed chromosomes, which activate very high levels of germline apoptosis in response to both the synapsis and the DNA damage checkpoints [64] (Fig 5A). *cmt-1* single mutants had wildtype levels of apoptosis (Fig 5B). In contrast to *syp-1* single mutants, *syp-1;cmt-1* double mutants had intermediate levels of germline apoptosis (Fig 5B), indicating that CMT-1 is required for either the synapsis or DNA damage checkpoint. To distinguish between these two checkpoints, we used *syp-1;spo-11* double mutants, which do not generate double strand breaks [18]. As a result, these double mutants do not activate the DNA damage checkpoint (Fig 5A) and produced an intermediate level of apoptosis (Fig 5B). [64]. When we generate *syp-1;spo-11;cmt-1*

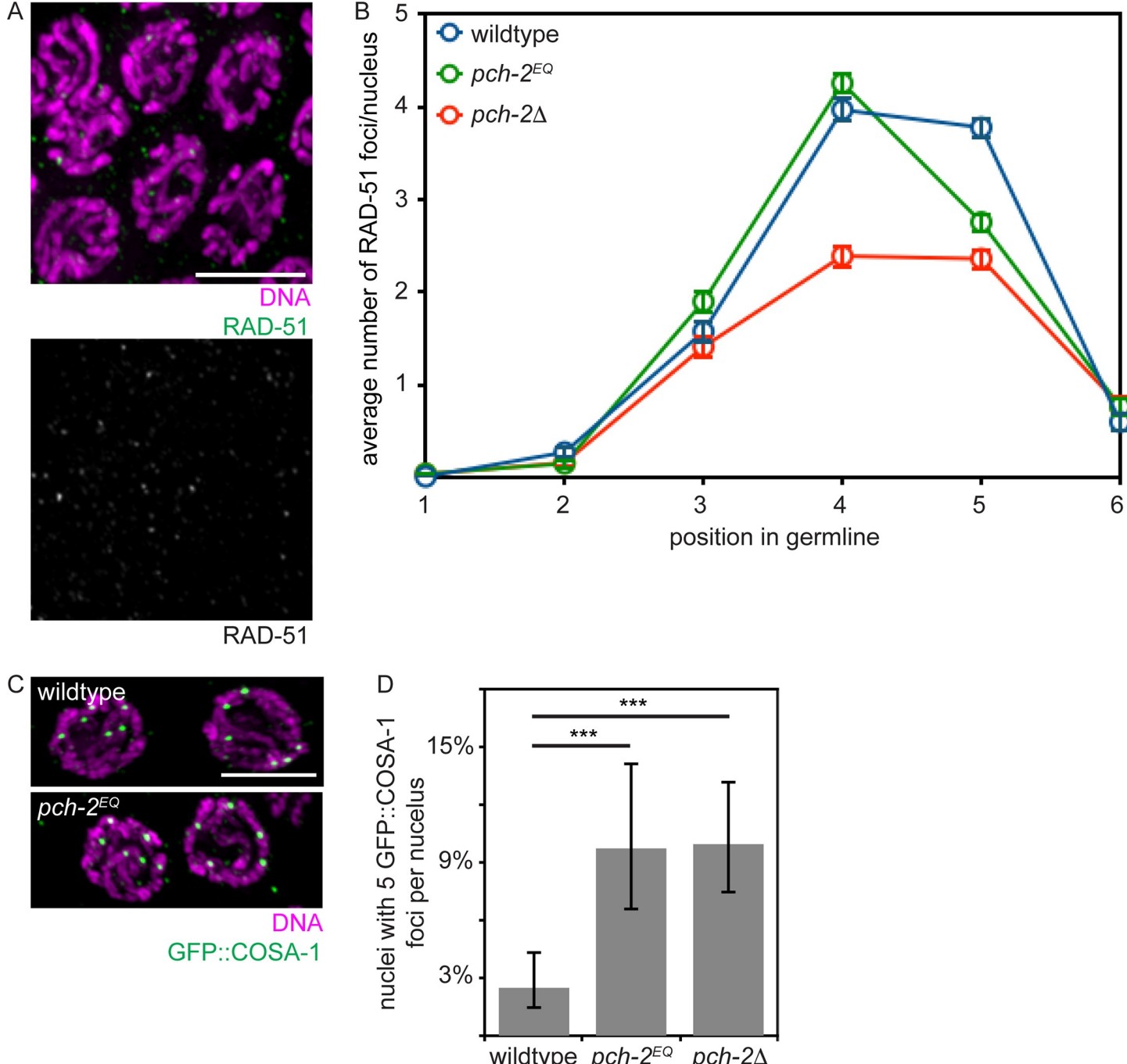

**Fig 4. Meiotic DNA repair is not affected in *pch-2^E253Q* mutants but crossover assurance is.** A. Images of meiotic nuclei stained with DAPI and antibodies against RAD-51. Color and grey scale images are provided. B. Timecourse of the average number of RAD-51 foci per nucleus in wildtype, *pch-2^E253Q* and *pch-2Δ* mutant germlines. Error bars indicate 2XSEM. C. Images of meiotic nuclei stained with DAPI and antibodies against GFP. D. Percentage of nuclei with five GFP::COSA-1 foci in wildtype animals, *pch-2^E253Q* and *pch-2Δ* mutants. Error bars indicate 95% confidence intervals. Significance was assessed by performing two-tailed Fisher exact tests. A *** indicates a p value < 0.0001.

triple mutants, we observed wildtype levels of apoptosis, similar to but slightly higher than *cmt-1* single mutants (Fig 5B) (p value < 0.05, two tailed t-test). Therefore, CMT-1 is required to activate apoptosis in response to the synapsis checkpoint, similar to *pch-2Δ* mutants.

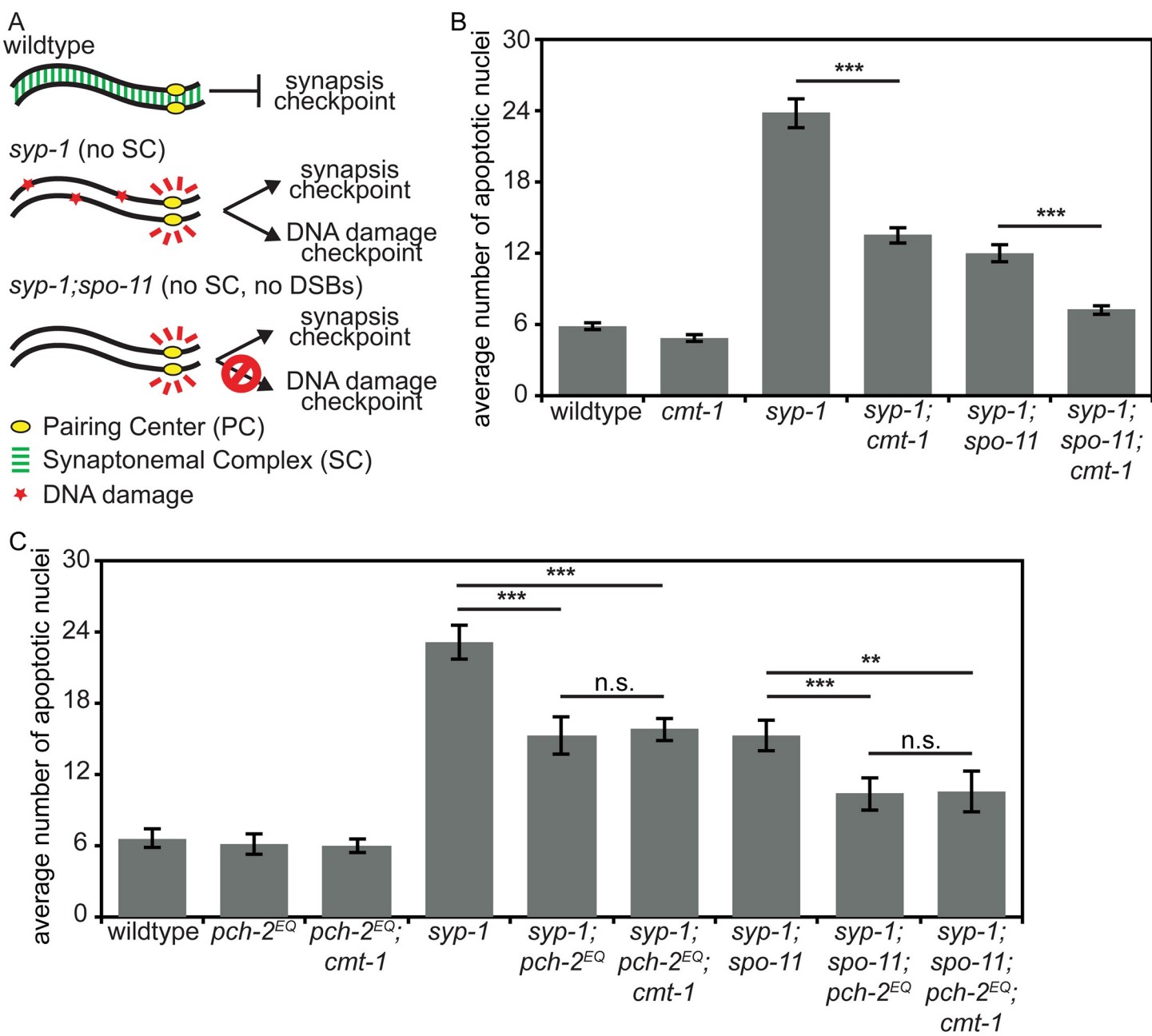

**Fig 5. CMT-1 is required for the synapsis checkpoint.** A. Cartoon of meiotic checkpoint activation in *C. elegans*. B. Mutation of *cmt-1* reduces apoptosis in *syp-1* single mutants and *syp-1;spo-11* double mutants. Error bars indicate 2XSEM. C. *pch-2^{E253Q}* reduces apoptosis in *syp-1* and *syp-1;spo-11* mutants but not in *syp-1;cmt-1* or *syp-1; spo-11;cmt-1* mutants. Significance was assessed by performing two-tailed t-tests. A *** indicates a p value < 0.0001, a ** indicates a p value < 0.01 and an n.s. indicates not significant.

## *cmt-1* mutants phenotypically resemble *pch-2^{E253Q}* mutants

Based on the requirement for CMT-1 in the synapsis checkpoint, we assessed pairing, synapsis, meiotic DNA repair and crossover formation in *cmt-1* mutants. Similar to our analysis of *pch-2^{E253Q}* mutants, pairing was delayed in *syp-1;cmt-1* double mutants (see zone 2, Fig 6A) (p value < 0.0001, two-tailed Fisher's exact test) and synapsis was accelerated in *cmt-1* single mutants (see zone 3, Fig 6B) (p value < 0.0001, two-tailed Fisher's exact test). We also detected

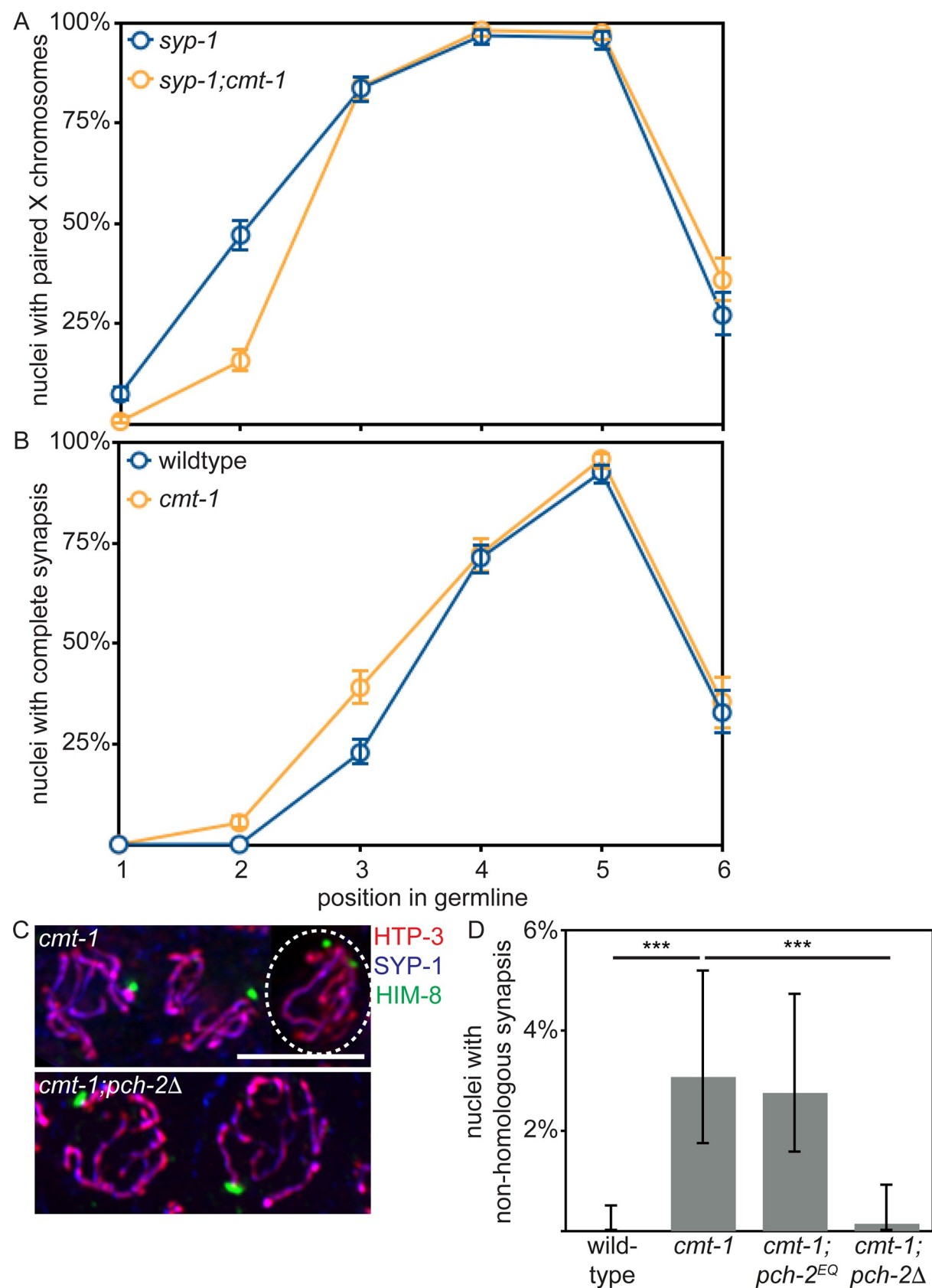

**Fig 6. *cmt-1* mutants delay pairing, accelerate synapsis and exhibit non-homologous synapsis, similar to *pch-2^E253Q^* mutants.** A. Timecourse of pairing in wildtype and *cmt-1* mutant germlines. B. Timecourse of synapsis in wildtype and *cmt-1* mutant germlines. C. Images of meiotic nuclei stained with antibodies against HTP-3, SYP-1 and HIM-8 in *cmt-1* and *cmt-1;pch-2Δ* mutants. The circled nucleus has undergone non-homologous synapsis. D. Quantification of non-homologous synapsis wildtype animals, *cmt-1*, *cmt-1;pch-2^E253Q^* and *cmt-1; pch-2Δ* mutants. All error bars indicate 95% confidence intervals. Significance was assessed by performing two-tailed Fisher exact tests. A *** indicates a p value < 0.0001.

non-homologous synapsis at levels similar to that of *pch-2^E253Q^* mutants (Figs 6C, 6D and S1). Meiotic DNA repair in *cmt-1* mutants, as visualized by the appearance and disappearance of RAD-51 (S3A Fig), and meiotic progression, as visualized by the appearance and disappearance of a protein required for DSB formation, DSB-1 (S2A and S2B Fig), closely resembled that of wildtype germlines. However, we did observe a delay in RAD-51 removal in *cmt-1* mutants (zones 5 and 6, p value < 0.0001, two-tailed Fisher's exact test). It is unclear whether this delay is biologically relevant, since we only observe a slight delay in DSB-1 removal (S2B Fig). In addition, crossover assurance was reduced (S3B Fig).

The meiotic phenotypes of *cmt-1* mutants are more similar to that of *pch-2^E253Q^* mutants than to that of *pch-2Δ* mutants, suggesting that PCH-2 can bind its meiotic substrates effectively in the absence of CMT-1 but CMT-1 is required for PCH-2's hydrolysis of ATP. This is in contrast to PCH-2's interaction with Mad2, which depends on CMT-1 [53, 61]. Consistent with this interpretation, *cmt-1;pch-2^E253Q^* double mutants had a similar frequency of non-homologous synapsis (Fig 6D), nuclei with five GFP::COSA-1 foci (S3B Fig), viable offspring and male-self progeny as either single mutant (Table 1). Finally, *syp-1;pch-2^E253Q^;cmt-1* and *syp-1;spo-11; pch-2^E253Q^;cmt-1* mutants did not exhibit any further reduction in germline apoptosis than *syp-1;pch-2^E253Q^* and *syp-1;spo-11;pch-2^E253Q^* mutants (Fig 5D) (p values = 0.596 and 0.891, respectively, two-tailed t-tests). Further, since *cmt-1* mutants result in meiotic phenotypes distinct from those we reported for spindle checkpoint mutants [65], specifically non-homologous synapsis, CMT-1's role in meiotic prophase is independent of its regulation of Mad2.

### Non-homologous synapsis in *cmt-1* mutants relies on PCH-2

We previously showed that CMT-1 is required for PCH-2 to localize to mitotic chromosomes during the spindle checkpoint [66]. We tested whether this was also true during meiotic prophase and found that CMT-1 was dispensable for PCH-2's localization to meiotic chromosomes (Fig 7A, grayscale images in S4 Fig). Consistent with CMT-1 being required for PCH-2's hydrolysis of ATP and release of substrates, we observed that PCH-2 stayed on meiotic chromosomes slightly longer than in wildtype germlines (Fig 7C).

During mitotic divisions in the *C. elegans* embryo, loss of CMT-1 partially suppresses the spindle checkpoint defect observed in *pch-2* mutants [66]. Since PCH-2 ensures availability of the inactive conformer of Mad2 so that it can be converted to the active one during checkpoint activation [67, 68], we hypothesize that this genetic interaction results because loss of CMT-1 makes more inactive Mad2 available. In this way, PCH-2 and CMT-1 would control the spindle checkpoint by regulating available, soluble pools of inactive and active Mad2. We

**Table 1. Viability and Him phenotype.**

| Genotype | % viability (total number of progeny) | % male self-progeny |
|---|---|---|
| wildtype | 100% (3949) | 0.05% |
| *pch-2^E253Q^* | 100% (2349) | 0.34% |
| *cmt-1* | 100% (3654) | 0.27% |
| *cmt-1;pch-2^E253Q^* | 100% (2721) | 0.25% |

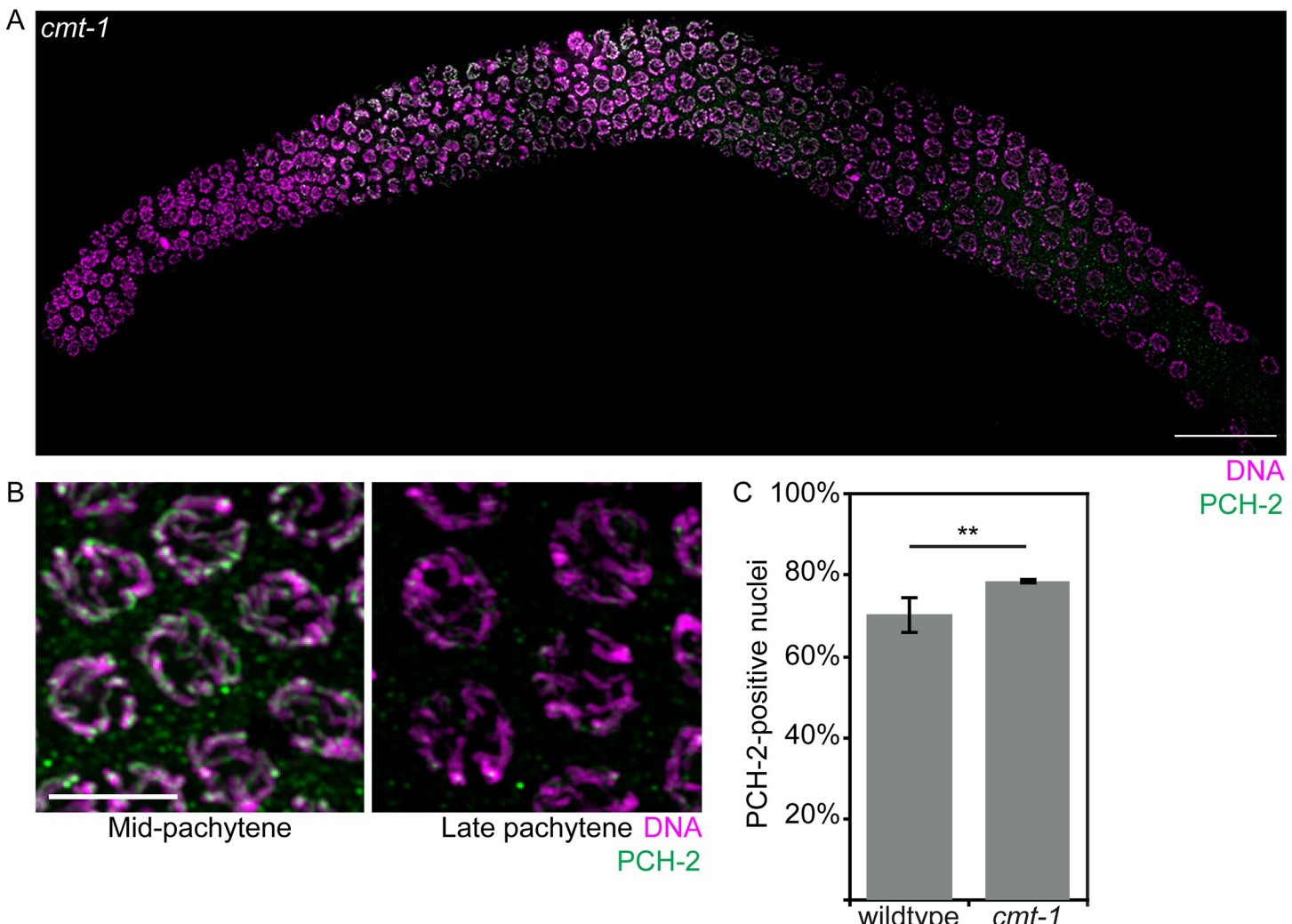

**Fig 7. PCH-2 localizes to meiotic chromosomes in *cmt-1* mutants.** A. Whole germline image of PCH-2 and DAPI staining in a *cmt-1* mutant germline. Scale bar indicates 20 microns. B. Meiotic nuclei in mid-pachytene stained with DAPI and antibodies against PCH-2 in *cmt-1* mutants. C. Quantification of percentage of PCH-2-positive nuclei in wildtype and *cmt-1* mutant germlines. Error bars indicate 95% confidence intervals. Significance was assessed by performing two-tailed t-tests. A ** indicates a p value < 0.01.

wondered if there was a similar relationship between PCH-2 and CMT-1 in meiotic prophase. To test this, we generated *cmt-1;pch-2Δ* double mutants and assayed non-homologous synapsis. Unlike *cmt-1* single mutants and similar to *pch-2Δ* mutants, we did not detect non-homologous synapsis in *cmt-1;pch-2Δ* double mutants (Fig 6C and 6D). Thus, PCH-2 is epistatic to CMT-1 during meiosis, at least in the context of non-homologous synapsis, supporting our interpretation that PCH-2 can bind its meiotic substrates independent of CMT-1 and suggesting that some of PCH-2's regulation of its meiotic substrates occurs directly on chromosomes.

## Discussion

*In vitro*, PCH-2$^{E253Q}$ protein binds ATP and its spindle checkpoint substrate, Mad2, but fails to remodel it, due to an inability to hydrolyze ATP [53]. Here, we show that the meiotic phenotypes observed in *pch-2$^{E253Q}$* mutants are consistent with a role for PCH-2 in disassembling inappropriate pairing, synapsis and crossover recombination intermediates in *C. elegans*,

resulting in non-homologous synapsis (Figs 3C, 3D and S1) and a loss of crossover assurance (Fig 4D). The effect we observe on these meiotic prophase events seems limited, affecting a small proportion of nuclei, but this may reflect either the inherent fidelity of these processes or the existence of redundant mechanisms that contribute to fidelity in *C. elegans*. Further, a role for PCH-2 in proofreading may be more apparent in organisms or situations in which homolog interactions are more challenging. Indeed, PCH-2's involvement in the interchromosomal effect in *Drosophila* [69], in which chromosome rearrangements affect crossover control on other chromosomes, supports this possibility. Nevertheless, our results demonstrate that PCH-2 proofreads interactions between homologous chromosomes to ensure that they are correct.

We previously showed that PCH-2 decelerates homolog pairing, synapsis and recombination, coordinating these meiotic prophase events [50]. Here, we argue that PCH-2 acts directly on chromosomes to proofread meiotic interhomolog interactions and ensure their fidelity. This role is supported by several pieces of data. First, PCH-2 localizes to meiotic chromosomes, both before and after synapsis [50]. This localization is functional, as demonstrated by its persistence on the synaptonemal complex and correlation with an extension of homolog access in mutants defective in recombination [50, 70]. Second, despite the defects we report, *pch-2Δ*, *pch-2^E253Q* and *cmt-1* mutants do not show acceleration of meiotic progression, as assayed by SUN-1 phosphorylation [50], appearance of transition zone nuclei (Figs 1 and 7 and [50]), ZIM-2 (S1 Fig) or DSB-1 localization (S2 Fig), indicating that PCH-2 and CMT-1 do not coordinate these meiotic prophase events indirectly through misregulation of meiotic progression. Finally, meiotic defects associated with loss of CMT-1 rely on PCH-2 function (Fig 6D), unlike what we observe in mitosis [66]. In mitosis, CMT-1 is epistatic to PCH-2, which we attribute to misregulation of soluble pools of Mad2. Soluble pools of at least one meiotic HORMAD, HTP-1, have been implicated in ensuring both accurate homolog synapsis and proper meiotic progression [23]. If PCH-2 and CMT-1 were regulating homolog pairing, synapsis and recombination via control of meiotic progression, we might predict that CMT-1 would be epistatic to PCH-2 in meiosis, as it is in mitosis. Taken together, these data favor a model in which PCH-2 and CMT-1 proofread interactions between homologous chromosomes that underlie pairing, synapsis and recombination in *C. elegans*. Since pairing, synapsis and recombination depend on meiotic HORMADs adopting their closed conformations [45], we hypothesize that these interactions between homologous chromosomes either involve or are stabilized by closed versions of meiotic HORMADs and PCH-2, in collaboration with CMT-1, remodels them to disassemble or destabilize them. In systems such as plants, yeast and mice, we propose that PCH-2 plays a similar role in addition to removing or relocalizing meiotic HORMADs to signal meiotic progression [28, 30–32].

We were surprised to see that PCH-2 localized to meiotic chromosomes in *cmt-1* mutants (Fig 7), unlike what we observe during the spindle checkpoint response [66], and that *cmt-1* mutants closely resembled *pch-2^E253Q* mutants in exhibiting non-homologous synapsis and a loss of crossover assurance (Figs 6 and S2). Since PCH-2's localization to meiotic chromosomes depends on synapsis [50], this localization may not accurately represent interaction with its proposed meiotic substrates, meiotic HORMADs. Alternatively, these data may suggest that PCH-2 is competent to bind meiotic HORMADs in the absence of CMT-1 and that CMT-1's meiotic function is to promote PCH-2's ATPase activity and ability to remodel these substrates. This is unlike its role in the spindle checkpoint, in which CMT-1/p31^comet is essential for PCH-2 to bind its substrate, Mad2 [53, 61, 62, 71]. This difference raises the possibility that the interaction between PCH-2 and meiotic HORMADs during their remodeling is substantially different than that with Mad2. This hypothesis is supported by the findings that budding yeast, in which *PCH2* was originally identified [72], does not appear to have a CMT-1/p31^comet ortholog [73, 74] and budding yeast Pch2 can directly interact with and remodel the budding yeast meiotic HORMAD, Hop1, *in vitro* [29]. However, another possibility is that PCH-2's ability to remodel meiotic

HORMADs exists in two modes: 1) a mode that proofreads meiotic interhomolog interactions during pairing, synapsis and recombination and depends on CMT-1 for ATP hydrolysis, but not substrate binding; and 2) one that removes meiotic HORMADs during meiotic progression which relies on CMT-1 for both substrate recognition and ATP hydrolysis. Such a model raises the immediate question of how these two modes might be regulated in an organism with both.

The PCH-2/HORMAD genetic module is evolutionarily ancient [75, 76], having been identified as operons in several bacteria [77], suggesting that this pair of proteins has been co-opted to function in multiple molecular contexts. While we can easily detect an interaction between PCH-2 and CMT-1, and CMT-1 and Mad2, by two-hybrid experiments [66], we have not been able to observe a similar interaction between PCH-2 or CMT-1 and any of the four meiotic HORMADs present in *C. elegans*, HTP-3, HIM-3, HTP-1 and HTP-2, making it difficult to identify what regions of these proteins are required to interact with PCH-2. Genetic mutations, combined with cytological analysis, seems the most straightforward way, at least in *C. elegans*, to understand whether and how PCH-2 and meiotic HORMADs interact to regulate meiotic pairing, synapsis and recombination.

## Materials and Methods

### Genetics and worm strains

The *C. elegans* Bristol N2 [78] was used as the wild-type strain. All strains were maintained at 20˚C under standard conditions unless otherwise stated. Mutations and rearrangements used were as follows:

LG I: *cmt-1(ok2879)*

LG II: *pch-2(blt5)*, *meIs8 [Ppie-1::GFP::cosa-1 + unc-119(+)]*

LG IV: *spo-11(ok79)*, *nT1[unc-?(n754) let-?(m435)] (IV, V), nTI [qIs51]*

LG V: *syp-1(me17)*, *bcIs39 [lim-7p::ced-1::GFP + lin-15(+)]*

The *pch-2(blt5)* allele, referred to as *pch-2$^{E253Q}$*, was created by CRISPR-mediated genomic editing as described in [51, 52]. pDD162 was mutagenized using Q5 mutagenesis (New England Biolabs) and oligos GTTTTTGTTCTTATCGACGGTTTTAGAGCTAGAAATAGC AAGT and CAAGACATCTCGCAATAGG. The resulting plasmid was sequenced and two different correct clones (50ng/ul total) were mixed with pRF4 (100ng/ul) and the repair oligo CTCGTTCAAAAAATGTTCGATCAAATTGATGAACTAGCTGAAGATGAGAAGTGCAT GGTTTTTGTGCTCATCGACCAAGTTTGATTTTTTTAAAAAACAATTTTTCTGGTTTT CATCAGTTTTTATGTCAGGTTGAAT (30ng/ul). Wildtype worms were picked as L4s, allowed to age 15–20 hours at 20˚C and injected with the described mix. Worms that produced rolling progeny were identified and F1 rollers, as well as their wildtype siblings, were placed on plates seeded with OP50, 1–2 rollers per plate and 6–8 non-rolling siblings per plate, and allowed to produce progeny. PCR and Bsp1286I digestions were performed on these F1s to identify worms that contained the mutant allele and individual F2s were picked to identify mutant homozygotes. Multiple homozygotes carrying the *pch-2(blt5)* mutant allele were backcrossed against wildtype worms at least three times and analyzed to determine whether they produced the same mutant phenotype.

### Antibodies, Immunostaining and Microscopy

DAPI staining and immunostaining was performed as in [64] 20 to 24 hours post L4 stage. Primary antibodies were as follows (dilutions are indicated in parentheses): rabbit anti-PCH-2

(1:500) [50], rabbit anti-SYP-1 (1:500) [6], chicken anti-HTP-3 (1:250) [54], guinea pig anti-HTP-3 (1:250) [54], rabbit anti-ZIM-1 (1:1000), guinea pig anti-ZIM-2 (1:2500) [56], guinea pig anti-HIM-8 (1:250) [55], mouse anti-GFP (1:100) (Invitrogen), guinea pig anti-DSB-1 (1:500) [60] and rabbit anti-RAD-51 (1:1000) (Novus Biologicals). Secondary antibodies were Cy3 anti-rabbit, anti-guinea pig and anti-chicken (Jackson Immunochemicals) and Alexa-Fluor 488 anti-guinea pig and anti-rabbit (Invitrogen). All secondary antibodies were used at a dilution of 1:500.

All images were acquired using a DeltaVision Personal DV system (Applied Precision) equipped with a 100X N.A. 1.40 oil-immersion objective (Olympus), resulting in an effective XY pixel spacing of 0.064 or 0.040 μm. Three-dimensional image stacks were collected at 0.2-μm Z-spacing and processed by constrained, iterative deconvolution. Image scaling and analysis were performed using functions in the softWoRx software package. Projections were calculated by a maximum intensity algorithm. Composite images were assembled and some false coloring was performed with Adobe Photoshop.

Scoring of germline apoptosis was performed as previously described in [64] in strains containing *bcIs39 [lim-7p::ced-1::GFP + lin-15(+)]* with the following exceptions. L4 hermaphrodites were allowed to age for 22 hours. They were then mounted under coverslips on 1.5% agarose pads containing 0.2mM levamisole and scored. A minimum of twenty-five germlines was analyzed for each genotype.

Quantification of pairing, synapsis, RAD-51 foci, GFP::COSA-1 foci, and DSB-1 positive nuclei was performed on animals 24 hours post L4 stage and with a minimum of three germlines per genotype. Relevant statistical analysis, as indicated in the Figure Legends, was used to assess significance. The number of nuclei assayed for each genotype for all figures is shown in S1 Table.

## Supporting information

**S1 Fig. Autosomes undergo non-homologous synapsis in *pch-2^E253Q* and *cmt-1* mutants.** A. Images of meiotic nuclei stained with antibodies against HTP-3, SYP-1 and ZIM-2 in wildtype animals, *pch-2^E253Q* and *cmt-1* mutants. Circled nuclei have undergone non-homologous synapsis. B. Quantification of non-homologous synapsis wildtype animals, *pch-2^E253Q* and *cmt-1* mutants. Error bars indicate 95% confidence intervals. Significance was assessed by performing two-tailed Fisher exact tests. A *** indicates a p value < 0.0001.
(TIF)

**S2 Fig. Meiotic progression is unaffected in *pch-2^E253Q* and *cmt-1* mutants.** A. Whole germline images of DSB-1 and DAPI staining in a wildtype, *pch-2^E253Q* and *cmt-1* mutant germline. Scale bar indicates 20 microns. B. Quantification of percentage of DSB-1-positive nuclei in wildtype, *pch-2^E253Q* and *cmt-1* mutant germlines. Error bars indicate 95% confidence intervals. Significance was assessed by performing two-tailed t-tests. A * indicates a p value < 0.05 and an n.s. indicates not significant.
(TIF)

**S3 Fig. Meiotic DNA repair is not affected in *cmt-1* mutants but crossover assurance is, similar to *pch-2^E253Q* mutants.** A. Timecourse of the average number of RAD-51 foci per nucleus in wildtype and *cmt-1* mutant germlines. Error bars indicate 2XSEM. B. Percentage of nuclei with five GFP::COSA-1 foci in wildtype animals, *cmt-1* single and *cmt-1;pch-2^E253Q* double mutants. Error bars indicate 95% confidence intervals. Significance was assessed by performing two-tailed Fisher exact tests. A *** indicates a p value < 0.0001.
(TIF)

**S4 Fig. Grayscale images of PCH-2 antibody staining in wildtype animals, *pch-2*[E253Q] and *cmt-1* mutants.** A. Whole germline images. Scale bar indicates 20 microns. B. Meiotic nuclei in mid-pachytene. C. Meiotic nuclei in late pachytene.
(TIF)

**S1 Table. Number of nuclei assayed for each genotype for all figures.**
(DOCX)

## Acknowledgments

We would like to thank Arshad Desai, Karen Oegema, Abby Dernburg, and Anne Villeneuve for valuable strains and reagents.

## Author Contributions

**Conceptualization:** Stefani Giacopazzi, Daniel Vong, Alice Devigne, Needhi Bhalla.

**Formal analysis:** Stefani Giacopazzi, Daniel Vong, Alice Devigne, Needhi Bhalla.

**Funding acquisition:** Needhi Bhalla.

**Investigation:** Stefani Giacopazzi, Daniel Vong, Alice Devigne, Needhi Bhalla.

**Methodology:** Stefani Giacopazzi, Daniel Vong, Alice Devigne, Needhi Bhalla.

**Project administration:** Needhi Bhalla.

**Supervision:** Needhi Bhalla.

**Writing – original draft:** Needhi Bhalla.

**Writing – review & editing:** Stefani Giacopazzi, Daniel Vong, Alice Devigne, Needhi Bhalla.

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
