## [Decision Letter · Decision Letter 0]

6 Feb 2020

Dear Dr Bhalla,

Thank you very much for submitting your Research Article entitled 'PCH-2 collaborates with CMT-1 to proofread meiotic homolog interactions.' to PLOS Genetics. Your manuscript was fully evaluated at the editorial level and by independent peer reviewers. The reviewers appreciated the attention to an important problem, but raised some substantial concerns about the current manuscript. Based on the reviews, we will not be able to accept this version of the manuscript, but we would be willing to review again a much-revised version. We cannot, of course, promise publication at that time.

Should you decide to revise the manuscript for further consideration here, your revisions should address the specific points made by each reviewer. Based on the reviews, we encourage you to include a more detailed quantification and analysis of both the pairing defects of the autosomes as well as the non-homologous synapsis, occurring in the mutants described in this manuscript. Some of these experiments were stated as data not shown in the manuscript but this data should be included in the manuscript figures.  We encourage you to pay particular attention to the excellent suggestions from Reviewers 2 and 3 to carry out these experiments.  Further, all three reviewers requested that the existing and additional pairing and non-homologous synapsis quantifications include the inclusion of specific mutants, such as cmt-1 and cmt-1; pch-2E253Q.  Importantly, statistics and n-values for all of your experiments should be included throughout the manuscript.  As mentioned by Reviewers 1 and 2, please consider revising the use of “proofreading” for describing the role of PCH-2.  We will also require a detailed list of your responses to the review comments and a description of the changes you have made in the manuscript.

If you decide to revise the manuscript for further consideration at PLOS Genetics, please aim to resubmit within the next 60 days, unless it will take extra time to address the concerns of the reviewers, in which case we would appreciate an expected resubmission date by email to plosgenetics@plos.org.

[LINK]

We are sorry that we cannot be more positive about your manuscript at this stage. Please do not hesitate to contact us if you have any concerns or questions.

Yours sincerely,

Diana E. Libuda, Ph.D.

Guest Editor

PLOS Genetics

Gregory Barsh

Editor-in-Chief

PLOS Genetics

Reviewer's Responses to Questions

**Comments to the Authors:**

Reviewer #1: In the manuscript by Giacopazzi, Vong et al, the authors characterize the in vivo phenotypes of a mutant in PCH-2 (pch-2E253Q). In vitro characterization previously showed that this mutant could bind its substrate but could not hydrolyze ATP. They find that pch-2E253Q mutants localize PCH-2E253Q but have a delay in pairing, accelerate synapsis, have a small fraction of non-homologous synapsis, and lose crossover assurance. The phenotype of non-homologous synapsis was different in the pch-2E253Q than in the pch-2 mutants, suggesting that the pch-2E253Q mutant is trapping an intermediate. They then ask if CMT-1 is needed for these events. They find that the phenotypes of cmt-1 mutants are similar to that of pch-2E253Q mutants, with a delay in pairing, accelerated synapsis, non-homologous synapsis, and reduced crossover assurance.

This manuscript details interesting results to test their hypothesis that they can reveal intermediates with the pch-2E253Q mutants and that the adaptor CMT-1 is not required for binding substrates but is required to remodel them. However, there are some further questions that need to be addressed:

1) In Figure 2C, the authors are arguing that they can trap the pairing intermediates in pch-2E253Q mutants. It would be more convincing if they directly show the difference between pch-2delta and pch-2E253Q, along with wildtype.

2) Similarly, the comparison of HIM-8 foci in pch-2E253Q and pch-2delta is needed to show that this is a defect specific to trapping intermediates.

3) What is the range of numbers and percent of each of GFP::COSA-1 foci? Having this range will help understand the severity of the defect.

4) Is there an increase in chromosome mis-segregation in the pch-2E253Q mutants, or less viable offspring?

5) Does the role of CMT-1 in activating apoptosis require PCH-2 and ATP hydrolysis? In the pch-2E253Q mutant, what are the levels of apoptosis and does the mutant decrease apoptosis in syp-1 mutants, like cmt-1 mutants do?

6) To be sure that CMT-1 and PCH-2 are acting together to disrupt non-homologous synapsis, the cmt-1; pch-2E253Q double mutant experiment should be performed to see if there is an additive effect or not.

7) I am struggling with the role of PCH-2 in proofreading versus a cell-cycle regulator needed to couple pairing and synapsis. I am wondering if the authors can discuss these differences further and focus on why they are suggesting proofreading?

Minor issues:

1) When referring to yeast proteins and genes, the yeast terminology should be used. Pch2 = protein, PCH2 = gene

2) In line 229-231, the authors conclude that PCH-2 can bind the meiotic substrates in the absence of CMT-1, but they do not show that until later. I found this statement confusing.

3) Line 149 get rid of comma after “in contrast”

4) Line 277-278 unneeded “localized”.

Reviewer #2: Review attached

Reviewer #3: The manuscript by Giacopazzi et al., provides insight into the role of the enigmatic AAA-ATPase PCH-2 in ensuring faithful meiotic prophase events. Taking advantage of a mutation in the conserved Walker B motif that is able to bind ATP but no hydrolyze it, the authors monitor localization, pairing, synapsis, and meiotic recombination progression. They find that pch-2E235Q has delayed pairing (of the X chromosome), faster SC formation leading to some nonhomologous synapsis, and relatively normal RAD-51 kinetics, but a slight reduction in the frequency of COSA-1 foci, suggesting a problem with crossover assurance. Most of these phenotypes are also observed in the p31comet ortholog, CMT-1; however, unlike its role in the spindle assembly checkpoint, CMT-1 is not required to localize PCH-2 to meiotic chromosomes. Together these results suggest that CMT-1 plays a distinct role in monitoring meiotic prophase events versus its canonical role in the spindle assembly checkpoint pathway. There are some interesting data presented, although not much insight into the underlying mechanism. In many figures, there are no statistics. Further, including the pch-2 null mutant in the analysis and reorganizing the figures would make it easier to compare pch-2 null, pch-2E235Q and cmt-1.

Abstract: “In vitro, this mutant binds its substrates . . .” – to my knowledge this has only been demonstrated for MAD-2 in the spindle assembly checkpoint and not for potential meiotic substrates (Ye, et al., 2015). This should be changed to reflect this. Further on line 29 in the abstract, please change “indicate” to “suggest” as no direct binding of meiotic substrates is shown in this manuscript.

I recommend that the data for the cmt-1 mutant be incorporated into the figures for the pch-2E235Q as indicated below.

Figure 1: Please indicate that this is antibody staining. There is significantly more background/non-nuclear staining in the mutant compared to wild type. Is this consistent over different experiments? Either replace the figure or suggest why it appears different. Please add the PCH-2 staining in cmt-1 mutant to this figure instead of figure 7.

Figure 2: As the X chromosome has different properties compared to the autosomes, it would be nice to include analysis of pairing of an autosome. Please also add the analysis of pairing of the pch-2 null and cmt-1 mutants in this figure (Fig 6A). Statistics should also be included.

Figure 3: Please quantify the extent of non-homologous synapsis. The data for ZIM-1 and ZIM-2 should be included, as well as the data for pch-2 null and cmt-1 mutants (Figure 6B).

Figure 4: Please add the pch-2 null and cmt-1 data from Supplemental Figure 2 here as well as statistics.

Minor: Several times throughout the manuscript meiotic DNA repair is written (line 98, line172, and others), I think is should be DSB. There is also a typo on line 278.

**Have all data underlying the figures and results presented in the manuscript been provided?**

Reviewer #1: None

Reviewer #2: Yes

Reviewer #3: No: The authors include "data not shown" - this is also indicated in my comments to the authors.

PLOS authors have the option to publish the peer review history of their article (what does this mean?). If published, this will include your full peer review and any attached files.

Reviewer #1: No

Reviewer #2: No

Reviewer #3: No

---

## [Decision Letter · Decision Letter 1]

1 Jun 2020

Dear Dr Bhalla,

We are pleased to inform you that your manuscript entitled "PCH-2 collaborates with CMT-1 to proofread meiotic homolog interactions." has been editorially accepted for publication in PLOS Genetics. Congratulations!

Yours sincerely,

Diana E. Libuda, Ph.D.

Guest Editor

PLOS Genetics

Gregory Barsh

Editor-in-Chief

PLOS Genetics

Comments from the reviewers (if applicable):

The reviewers all agree the manuscript is ready to be accepted, but given that Reviewer #2 noted a desire for more extensive analysis of non-homologous synapsis and Reviewer #3 requested a specific wording change in the abstract with regards to the extensiveness of the defects on both non-homologous synapsis and crossover assurance, the author must make the Reviewer #3 requested wording change in the abstract prior to publication.  In addition, the authors should make the Reviewer #2 requested changes to the supplemental figures.

Reviewer's Responses to Questions

**Comments to the Authors:**

Reviewer #1: The authors have appropriately revised the manuscript and addressed all of my concerns. I am happy with the changes.

Reviewer #2: The revised version of the manuscript “PCH-2 collaborates with CMT-1 to proofread meiotic homolog interactions,” is much improved and now acceptable for publication. The authors completed the majority of the revisions and have reasonable responses to the ones not completed. While I still believe it is worth expanding the examination of X and autosome pairing and the non-homologous synapsis phenotype, the manuscript stands without it. Some minor issues:

The Fig S4 figure legend should say that it is antibody staining against PCH-2.

While this is likely a compiling issue the first two supplemental figures in the PDF are the old Fig S1 and Fig S2.

Reviewer #3: The revised manuscript by Giacopazzi et al., provides insight into the role of the enigmatic AAA-ATPase PCH-2 in ensuring faithful meiotic prophase events. The authors have done a good job addressing the previous reviews, and the resulting manuscript is easier to follow and provides important information about the role of PCH-2 and CMT-1 in meiotic prophase. I have two minor suggestions:

1. In the Abstract, I recommend changing: “this mutation results in non-homologous synapsis and loss of crossover assurance” to “this mutation results in some non-homologous synapsis and impaired crossover assurance”

2. I ask the authors to consider the tense in the results section.

**Have all data underlying the figures and results presented in the manuscript been provided?**

Reviewer #1: Yes

Reviewer #2: Yes

Reviewer #3: Yes

PLOS authors have the option to publish the peer review history of their article (what does this mean?). If published, this will include your full peer review and any attached files.

Reviewer #1: No

Reviewer #2: No

Reviewer #3: No

**Data Deposition**

http://datadryad.org/submit?journalID=pgenetics&manu=PGENETICS-D-20-00029R1

**Press Queries**

---

## [Editor Report · Acceptance letter]

22 Jul 2020

PGENETICS-D-20-00029R1 

PCH-2 collaborates with CMT-1 to proofread meiotic homolog interactions. 

Dear Dr Bhalla, 

We are pleased to inform you that your manuscript entitled "PCH-2 collaborates with CMT-1 to proofread meiotic homolog interactions." has been formally accepted for publication in PLOS Genetics! Your manuscript is now with our production department and you will be notified of the publication date in due course.

With kind regards,

Kaitlin Butler

PLOS Genetics

On behalf of:
